# Role of Mitochondrial DNA Damage in ROS-Mediated Pathogenesis of Age-Related Macular Degeneration (AMD)

**DOI:** 10.3390/ijms20102374

**Published:** 2019-05-14

**Authors:** Kai Kaarniranta, Elzbieta Pawlowska, Joanna Szczepanska, Aleksandra Jablkowska, Janusz Blasiak

**Affiliations:** 1Department of Ophthalmology, University of Eastern Finland, 70211 Kuopio, Finland; kai.kaarniranta@kuh.fi; 2Department of Ophthalmology, Kuopio University Hospital, 70029 Kuopio, Finland; 3Department of Orthodontics, Medical University of Lodz, 92-216 Lodz, Poland; elzbieta.pawlowska@umed.lodz.pl; 4Department of Pediatric Dentistry, Medical University of Lodz, 92-216 Lodz, Poland; joanna.szczepanska@umed.lodz.pl; 5Department of Infectious and Liver Diseases, W. Bieganski Hospital, 91-347 Lodz, Poland; jablkowska@pro.onet.pl; 6Department of Molecular Genetics, Faculty of Biology and Environmental Protection, University of Lodz, 90-236 Lodz, Poland

**Keywords:** age-related macular degeneration, mitochondria, mtDNA damage, DNA damage response, reactive oxygen species

## Abstract

Age-related macular degeneration (AMD) is a complex eye disease that affects millions of people worldwide and is the main reason for legal blindness and vision loss in the elderly in developed countries. Although the cause of AMD pathogenesis is not known, oxidative stress-related damage to retinal pigment epithelium (RPE) is considered an early event in AMD induction. However, the precise cause of such damage and of the induction of oxidative stress, including related oxidative effects occurring in RPE and the onset and progression of AMD, are not well understood. Many results point to mitochondria as a source of elevated levels of reactive oxygen species (ROS) in AMD. This ROS increase can be associated with aging and effects induced by other AMD risk factors and is correlated with damage to mitochondrial DNA. Therefore, mitochondrial DNA (mtDNA) damage can be an essential element of AMD pathogenesis. This is supported by many studies that show a greater susceptibility of mtDNA than nuclear DNA to DNA-damaging agents in AMD. Therefore, the mitochondrial DNA damage reaction (mtDDR) is important in AMD prevention and in slowing down its progression as is ROS-targeting AMD therapy. However, we know far less about mtDNA than its nuclear counterparts. Further research should measure DNA damage in order to compare it in mitochondria and the nucleus, as current methods have serious disadvantages.

## 1. Introduction

Reactive oxygen species (ROS), including free radicals, play important roles in cellular signaling, being an important element of organismal homeostasis [1]. On the other hand, ROS are implicated in the pathogenesis of many human diseases, and, in fact, it is not easy to find a disorder without ROS in its pathogenesis. Moreover, ROS are directly or indirectly implicated in both normal (physiological) and accelerated aging [2]. Therefore, it is not surprising that ROS are reported to play an important role in the etiology of several age-related diseases [3].

There are many sources of ROS in the cell, including enzymes of cytochrome P450 and other enzymes as well as the mitochondrial electron transport chain (mtETC) [4]. The latter is especially significant for several reasons—it produces ROS in its normal functioning, and the amount of ROS may increase greatly in malfunctioned mtETC, which in turn increases total ROS level (mitochondrial vicious cycle) [5]. In this way, mitochondria can be implicated in the process of aging, both normal and accelerated, as they involve ROS accumulation [6]. Moreover, mitochondrial reactive oxygen species (mtROS) are involved in the regulation of several other physiological and pathological processes [7].

The involvement of mtROS in the pathogenesis of many diseases has led to the idea of targeting them in therapy of various disorders [8,9,10,11]. It seems that age-related diseases are especially suited for this, as they are associated with mitochondria through the process of aging. However, at present, we do not know exactly many fundamental aspects of the involvement of mtROS in pathogenesis of age-related diseases. First, the cause of production and disposition of ROS by mitochondria is not completely known. Similarly, the question of how increased ROS levels induce pathophysiological events involved in a disease onset and/or progression still needs answering. The next question is: What element in mitochondria is primarily responsible for pathogenic ROS overproduction? In this review, we address these problems in relation to age-related macular degeneration (AMD).

## 2. Age-Related Macular Degeneration—An Eye Disease with the Critical Role of ROS in Its Pathogenesis

Age-related macular degeneration is a complex, progressive eye disease which is the main reason for legal blindness and vision loss in the elderly worldwide [12]. The estimated global pooled prevalence of AMD in 2013 was about 17%, and the number of individuals affected by AMD is 196 million and projected to increase to 288 million in 2040 [13,14]. The global costs of AMD are €101.1 million in the UK, €60.5 million in Italy, €91.4 million in Germany, and €51.3 million in France [15,16,17]. Therefore, the burden of AMD is an emerging element of global vision loss.

Despite the prevalence and high cost of medical care, available therapeutic options for AMD are very limited. This is likely due to the complexity of the disease and incomplete knowledge of the mechanisms underlying its pathogenesis. Therefore, studies on molecular aspects of AMD are justified and needed. However, such studies encounter major problems. First, molecular studies in live human subjects are limited, and postmortem research may add some misinformation. Second, animal models of the disease are often criticized as inadequate due to a substantial difference between human and rodent retinas [18]. Third, various cellular models of AMD may represent some features which cannot be observed in live retinas, such as retinal pigment epithelium (RPE) cells that do not proliferate in situ due to spatial constraints and other limitations.

The main clinical symptom of AMD is the impairment of central vision, which may eventually result in complete vision loss (Figure 1). Chronologically, AMD can be categorized as early and late. The early AMD is typified by the presence of and increase in deposits of extracellular debris between Bruch’s membrane and RPE. These debris are called drusen, and their presence emerges with AMD progression [19]. Late AMD may be manifested in two forms, atrophic (dry) and neovascular (wet). The former is characterized by the development of geographic atrophy (GA) and worsening of central vision. There is not any efficient treatment for dry AMD. Wet AMD causes more rapid and pronounced changes than observed in dry AMD and is characterized by choroidal neovascularization (CNV). New blood vessels often leak into the retina and cause hemorrhage, retinal detachment, and disciform scars. Currently, wet AMD is treated with repeated intravitreous injections of anti-vascular endothelial growth factor A (VEGFA) [20].

It is hypothesized and supported by many studies that the pathogenesis of AMD starts in the RPE, a single layer of cells located between the neuroretina and Bruch’s membrane [21]. The RPE plays the crucial functions in the maintaining of retinal homeostasis [22]. Age-related macular degeneration is a complex disease, and for the majority of complex traits, the mechanisms underlying their pathogenesis are not exactly known [23]. These mechanisms likely include interactions between genetic, environmental and lifestyle risk factors that may lead to aberrant processes occurring in the retina [24]. Advanced age is by definition the main AMD risk factor. The main genetic factors are associated with several loci, including histocompatibility locus antigen (HLA) and the alternate complement pathway—*CFH*, *CFB*, *CFI* and *C3*, as well as the *C2* and *ARMS2/HTRA1* gene regions [25,26]. Other frequently questioned risk factors are female sex, white race, smoking, diet rich in polyunsaturated acids and blue light exposure [26]. Damage to RPE is often observed in early stages of AMD, although in some cases it may be preceded by photoreceptor loss [27,28]. Many experiments and clinically relevant data support that RPE damage is directly or indirectly caused by oxidative stress [29,30,31,32]. It is not easy to determine the source of this stress, as most, if not all, AMD risk factors can be associated with overproduction of ROS, elevated levels of which are observed in oxidative stress.

Many studies performed on human subjects, experimental animals, and cell cultures clearly show oxidative damage to RPE and choriocapillaris. However, the precise mechanism of RPE damage, source of elevated levels of ROS and, most importantly, the exact association between oxidative effects occurring in RPE and the onset and progression of AMD still need explanation [33]. In this review, we present arguments that mitochondrial dysfunction underlined by damage to mitochondrial DNA (mtDNA) may be the reason for increased ROS production in RPE associated with AMD onset and progression. This concept is not entirely new, but we present certain novel arguments and update some previous data.

## 3. Mitochondria—A Central Structure in AMD Pathogenesis

Retinal pigment epithelium cells in the central retina do not proliferate in situ due to spatial constraints, and adult stem cells have not been identified in that structure. Therefore, if some of the cells in the central retina are damaged, they can be replaced by cells from the periphery that can proliferate, as they are more relaxed than cells in the central region. However, if these peripheral cells are affected by stress-induced senescence, this mechanism fails [34]. The closer to the center damaged cells are located, the less chance there is to replace them by proliferating cells from the periphery. This mechanism may indicate cellular senescence as a major effect associated with atrophy of the retina observed in dry AMD and may explain, at least in part, why the macula and not the peripheral retina is prone to degeneration by AMD pathogenesis factors. Dieguez et al. created a dry AMD model by superior cervical ganglionectomy in C57BL/6J mice to explain such localized susceptibility to AMD [35]. This model limits the area of the occurrence of effects to the temporal region of the RPE/outer retina. These authors found that the temporal region was characterized by a lower melanin content, thicker basal infoldings, higher mitochondrial mass, and higher levels of antioxidant enzymes as compared with its nasal counterpart. Superior cervical ganglionectomy resulted in a lower efficacy of the antioxidant system and a lower mass of mitochondria. Damage to mitochondria was observed exclusively in the temporal region of RPE. This model was created not to explore the role of mitochondria in AMD pathogenesis but to study its general mechanisms. However, it unequivocally indicates that mitochondria are main players in AMD pathophysiology. Moreover, damage to mitochondria in AMD may contribute to topological features of the disease, in particular its macular localization.

Zhao et al. showed that postnatal inhibition of oxidative phosphorylation in mice RPE mitochondria resulted in many mechanistic targets for rapamycin (mTOR) changes typical for degenerated RPE in AMD [36]. Feher et al. observed a decrease in the number and electron microscopy area of mitochondria and loss of cristae and matrix density in aging human RPE, but these changes were more pronounced in individuals with AMD than in subjects free of this disease [37]. These studies revealed that the extent of morphological changes in RPE observed in AMD could be reached in non-AMD patients after 10–15 years since their first appearance, suggesting that AMD may be characterized by accelerated aging.

A proteomic analysis performed in the Ferrington laboratory revealed changes in the expression of proteins involved in mitochondrial refolding and trafficking in RPE of AMD patients as compared with non-AMD subjects [38]. In a subsequent work from that laboratory, an alternated expression of the α-, β- and δ-subunits of the catalytic portion of ATP synthase, subunit VIb of the cytochrome *c* oxidase complex, mitofilin, mtHsp70, and the mitochondrial translation factor Tu was observed [39]. A positive correlation between changes in expression of these genes and the progressive stage of AMD was observed.

Ferrington et al. used primary RPE cells obtained from AMD donors and control subjects within 24 h of death [40]. They observed that AMD RPE had lower respiration and ATP production than controls. However, treatment with a high concentration of hydrogen peroxide did not affect ATP production in AMD patients, in contrast to controls, who displayed about a one-third decrease in ATP synthesis, suggesting a resistance to oxidative stress in RPE cells from AMD donors. This was also supported by an even more pronounced decrease in maximal respiration and spare capacity in control than AMD samples. Moreover, AMD cells were more resistant to oxidant-induced death. No difference in mtDNA content was observed, but RPE cells from AMD donors had higher levels of the transcriptional coactivator, peroxisome proliferator-activated receptor gamma coactivator 1-alpha (PGC-1α). In general, a higher activity of PGC-1α is associated with better mitochondrial functions [41]. The authors did not provide convincing arguments to explain this discrepancy. That work showed for the first time that AMD may be associated with an energy crisis located in mitochondria. Therefore, mitochondrial dysfunction could play a central role in AMD pathogenesis. This role is closely associated with ROS overproduction, which can be considered as both cause and consequence of mitochondrial dysfunction. Increased PGC-1α levels can result from challenging mitochondrial quality control to face oxidative stress. The resistance to oxidative stress in AMD is in an apparent conflict with the causative role of the stress in AMD pathogenesis and adaptive mechanisms. Ferrington et al. concluded that the cells they used, i.e., cultures of primary RPE cells isolated from AMD subjects, were a good cellular model to study AMD pathogenesis.

Prior to Ferrington’s paper, Golestaneh et al. showed that the susceptibility of cultured human RPE cells obtained from AMD donors did not differ from cells obtained from donors without AMD after a 24 h incubation with hydrogen peroxide, but after 48 h incubation, RPE cells from AMD donors were more susceptible to oxidative stress-induced cell death [42]. These authors also showed that mitochondria of RPE cells from AMD donors were dysfunctional, producing lower levels of ATP, whereas the ATP produced by glycolysis was higher, suggesting that ATP was essentially produced by glycolysis rather than mitochondrial activity, further supporting the hypothesis of the significance of energy crisis allocated to mitochondria in AMD pathogenesis.

## 4. Generation and Regulation of ROS by Mitochondria

Reactive oxygen species are continuously produced in the cell during its normal metabolism and play an important role in intracellular signaling. Normal level of ROS is controlled by the cellular antioxidant system, containing antioxidant enzymes, small molecular weight antioxidants, and DNA repair proteins. However, in some intra- and extracellular conditions, ROS level can surpass the antioxidant potential of the cells, leading to the state described as oxidative stress. Mitochondrial electron transport chain (mtETC), along with cytochrome P450, nicotinamide adenine dinucleotide phosphate (NADPH) oxidase (NOX), and xanthine oxidase (XO) are major sources of intracellular ROS [43]. Mitochondrial ROS seem to be of particular interest in the field of intracellular ROS metabolism and signaling, as the antioxidant system in mitochondria is far less known than its counterparts in the rest of the cell.

The mitochondrial electron transport chain produces ROS during its normal functioning, and they play an important role in cellular signaling [1,44]. However, the extent of ROS produced by mtETC may increase during mtETC malfunctions. These extra ROS are mainly produced by the I and III complexes of mtETC in coupling with induced proton leak [45]. A small imbalance in mtETC functions may lead to a transient accumulation of ROS, which could damage mtDNA in genes encoding mtETC components. Expression of these damaged genes may lead to synthesis of malfunctioned proteins of mtETC, further accumulation of ROS, and even more massive damage to mtDNA, leading in turn to the synthesis of faulty mtETC proteins and further ROS overproduction in repeated cycles [46]. This state is referred to as “mitochondrial vicious cycle”, which is considered as an important element of normal and premature aging [5,47]. However, mtDNA damage does not always result in increased ROS production [48]. Moreover, the sites of ROS production and mtDNA location, which is principally attached to the matrix side of the inner mitochondrial membrane, overlap, and so mtDNA must be repaired under ROS “bombardment”, affecting DNA repair proteins and lowering the efficacy of DNA repair [49]. Moreover, such a situation creates an opportunity to form mtDNA-protein crosslinks mediated by ROS, which are one of, if not the most serious form of DNA damage [50]. Therefore, the maintenance of mtDNA can be crucial to the proper functioning of mitochondria and can play an important role in the pathogenesis of mitochondria- or ROS-related diseases. Unrepaired or misrepaired damage to mtDNA may contribute to aging, so DNA damage response in mitochondria can also be important in age-related diseases.

## 5. DNA Damage Response in mtDNA

Human mtDNA is a double-stranded, closed DNA with 16,569 base pairs (bps). It is usually referred to as “circular”, although the probability of adopting the structure of a perfect circle by mtDNA is negligibly low. However, diagramming mtDNA as a circle is useful for presenting and analyzing its structure and function. In reality, mtDNA may adopt complex structures with supercoils and interwounds. Mitochondrial DNA, contrary to its nuclear counterpart, contains almost exclusively coding sequences. It has genes coding for 13 polypeptides that are all components of mtETC and several functional RNA species. The major non-coding region of mtDNA, the mitochondrial control region (CR), is important for mtDNA replication and transcription and has the highest mutational rate in mtDNA, which has not been exactly determined, which may follow from yet unrecognized source(s) of such hypervariability in CR [51]. One of the reasons for a high mutation rate in CR is the presence of a replication initiation site (origin) for the heavy mtDNA strand, which is denaturated in each replication cycle and prone to DNA-damaging factors.

There are many deletions and point mutations in mtDNA, and some of them are associated with serious human disorders, such as ophthalmoplegia, migraine, dysphagia, sensorineural hearing loss, cognitive decline, and others [52,53]. Each nucleated human cell may contain many molecules of mtDNA of different sequences, which leads to the state referred to as “heteroplasmy.” In this state, mutated mtDNAs coexist with their normal counterparts, and usually a pathogenic mutation must occur at a level high enough to contribute to a pathological phenotype—in several cases that level is determined as 85% [54].

Phenotypic consequences of mtDNA damage are determined by several factors, including the number of affected mitochondria, environmental conditions, and mechanisms of mtDNA maintenance [55]. There is no reason to state that mtDNA itself, i.e., as a chemical molecule, is differently susceptible to DNA damage than its nuclear counterpart. However, the subcellular localization and organization of mtDNA are substantially different than nuclear DNA (nDNA), which, along with different DNA damage responses in mitochondria and the nucleus, determines differences in the DNA damage spectrum between mitochondrial and nuclear DNAs. These differences also influence the precision of measurements of mtDNA damage, which, in general, is lower than nDNA.

Environmental damage to mtDNA is induced by essentially the same factors as in nDNA, but they may present different mechanisms of action. The main reason for this is the different metabolism of these factors and their intermediate products in these two organelles [46]. The main difference between mtDNA and nDNA damages arises from the exposure to endogenous factors. Due to the close proximity of mtETC, mtDNA is prone to oxidative damage, which may take the form of small modifications to the nitrogen bases and the deoxyribose ring, apurinic/apyrimidinic (AP) sites, strand breaks, chemical adducts of bases, and others [56]. Hydrogen peroxide, a frequently used inducer of oxidative stress, induces mainly AP sites in human cell cultures [57]. Apurinic/apyrimidinic sites can be converted to single-strand breaks (SSBs) and together can be the principal form of mtDNA damage [58]. Moreover, damage to the genes encoding mtETC components results in dysfunction of these components, leading to increased ROS production by mtETC, which may induce further damage to these genes—the mitochondrial vicious cycle [5]. Mitochondrial metabolism and the composition of the mitochondrial membrane underlie the production of reactive aldehydes in mitochondria, which may contribute to the formation of mtDNA adducts [59]. Damages specific to mtDNA result mainly from specificity of mitochondrial systems to deal with them. Base excision repair (BER) operates quite efficiently in mitochondria, and it can remove 8-oxo-7,8-dihydro-2′-deoxyguanosine (8-oxoG), a major oxidative modification of mtDNA, but if it fails, 8-oxoG can be further oxidized to produce its more stable and mutagenic forms, which may interfere with DNA replication [60,61]. Other aspects of mtDNA damage following its metabolism are presented in the next section.

Similar to the nucleus, damage to mtDNA may affect its replication and transcription as well as expression of mitochondrial genes. If non-repaired or misrepaired, mtDNA damage may turn into mutation, which can be maternally inherited.

DNA damage response (DDR) is an evolutionary reaction to DNA damage, which may interfere with the process of sending genetic information from one generation to the next. DDR has been recognized as less efficient in mitochondria than in the nucleus, contributing to a higher mutation rate in mtDNA than in nDNA [62,63]. DDR in mitochondria (mtDDR) is coupled with mitochondrial quality control (mtQC), a system responsible for mitochondria’s proper maintenance and functioning, including mitochondrial biogenesis and mitophagy [64]. One of the nuclear DDR pathways—apoptosis—is regulated by a controlled release of cytochrome *c* from mitochondria [65].

Similar to the nucleus, DNA repair is the main reaction in mtDDR (Figure 2). Initially, the mitochondrial DNA repair system was considered much poorer than its nuclear counterpart. At present, an emerging similarity between these two DNA repair systems has been recognized, which is supported by the recent discoveries in that field [66]. First, mtDNA is not so “naked” as it used to be believed because it is associated with several proteins involved in its replication, transcription, and maintenance. They are: Polymerase gamma (PolG, the mitochondrial replicase); single-stranded DNA binding protein 1 (mtSSB); twinkle mtDNA helicase; transcription factor A, mitochondrial (TFAM); prohibitin (PHB). Other proteins are added to that list to form a structure called nucleoid [67]. Therefore, lack of association with proteins is no longer an argument supporting a higher rate of mutations in mtDNA than in nDNA. Furthermore, that association is itself a controversial issue in mutagenesis, as DNA-damaging factors, usually a quantum of radiation or a molecule of a chemical compound, are small in size and can find their way even to DNA tightly associated with proteins. On the other hand, when highly organized DNA is damaged, its repair may be difficult as relatively large DNA repair proteins do not have direct access to the sites of damage.

Although mtDNA is much smaller than nDNA, the mechanism of its replication is poorly understood, and ribonucleotides can be present in mtDNA after it completes its replication [68]. Polymerase gamma has a proofreading activity and is supported by the action of PrimPol (primase and DNA-directed polymerase), providing primers for PolG, playing a role in mtDNA damage tolerance, and being required for reinitiation of replication stalled by mtDNA damage [69]. There are two distinct features of mtDNA maintenance distinguishing it from nDNA. First, highly damaged mtDNA can be degraded as there is no reason for the cell to stop the cell cycle and begin apoptosis, as can be observed in the nucleus when the extent of DNA damage exceeds the cell repair capacity. However, the precise mechanisms of mtDNA degradation is not known, and mitophagy, mitochondrial endonucleases, and PolG can be involved [70]. Moreover, a not only extensive but also small, yet persistent, damage to mtDNA may induce its degradation [48]. The imbalance in the number of mtDNA copies may be associated with pathogenesis of several mitochondrial diseases [71]. Second, nucleotide excision repair (NER), the most versatile DNA repair pathway, has not been proven to act in mitochondria, although the presence of some proteins playing a role in a form of nuclear NER has been observed in mitochondria [72,73]. Base excision repair acts in short- and long-path modes in mitochondria [74]. This is probably underlined by the great number of oxidative modifications to mtDNA bases resulting from a high concentration of ROS in mitochondria. Alternative excision repair (AER) may partly compensate the lack of removal of UV-induced DNA damage by NER in mitochondrial DNA, as shown in yeast [75]. DNA double-strand breaks (DSBs) belonging to the most serious DNA lesions, can, similar to the nucleus, be mainly repaired by homologous recombination repair (HRR) and non-homologous end joining (NHEJ), but the assortment of proteins involved in these pathways and mechanisms of their action can be different in mitochondria compared to the nucleus [76,77]. Moreover, microhomology-mediated NHEJ, which can also be seen as a functional variant of HRR, seems to dominate in DSB repair in mitochondria. Due to high concentration of ROS, DNA repair enzymes may be crosslinked with mtDNA, which is the case for PolG acting on AP sites oxidized at C1’ [78]. The crosslink can be induced by 2-deoxyribonolactone (dL), a product of the attack of hydroxyl radical on C1’ in a mtDNA nucleotide. Obviously, this is not the only possibility of forming mtDNA-protein crosslinks when attempting to repair damage to mtDNA. This situation is not very specific for mtDNA, because nDNA can also be crosslinked with DNA repair proteins. However, the mechanism of repairing such crosslinks in mitochondria is poorly understood and is expected to be less effective than in the nucleus. This problem has been recently reviewed by Caston and Demple [50].

If DNA damage in the nucleus cannot be repaired before replication or mitosis, the cell cycle is stopped to give the cell more time to repair, and it may activate a programmed death pathway or damage can be tolerated. Translesion synthesis (TLS) is a pathway enabling the cell to replicate its DNA despite damage, and it is a major mechanism of DNA damage tolerance, an important component of DDR [79]. There are TLS polymerases that specialize in bypassing DNA damage and DNA synthesis beyond damage—“extenders” and “inserters”, respectively. So far, such DNA polymerases have not been identified in mitochondria but, somewhat surprisingly, it has been suggested that TLS-like mechanisms in mitochondria are associated with PolG, the mitochondrial replicase [80]. As stated above, PolG can be supported in its action by PrimPol [68].

## 6. mtDNA Damage and Repair in AMD

Accumulation of mtDNA damage in mitochondria may result from several mechanisms, listed in the previous sections. This process can be associated with both normal and accelerated aging as well as several other pathological conditions [81]. Accumulation of the common 4977 bp deletion in mtDNA (ΔmtDNA 4977) was observed in aging but not in the fetal human RPE and neural retina [82]. Therefore, aging in the retina is linked with accumulation of mutations in mtDNA, leading to increased instability of mtDNA and dysfunctions of mitochondria (vicious cycle) and the retina.

Comparative analysis of variation in mtDNA in different tissues of AMD patients and non-AMD controls was performed in several studies. Jones et al. found that the mitochondrial haplogroup H was protective against DNA and soft drusen development, whereas the U haplogroup was associated with pronounced general detrimental changes in RPE [83]. Those studies were performed on blood obtained from a cohort of AMD patients and controls enrolled in the Blue Mountains Eye Study [84]. Udar et al. studied mtDNA haplogroups in the retinas of 10 AMD patients and 11 control subjects [85]. They found that the mt1626T>C and mt73A>G of the J and T haplogroups single-nucleotide polymorphisms (SNPs) occurred more frequently in retinas of AMD patients than controls. This association was confirmed in the blood of 99 AMD patients and 92 controls. Cantar et al. observed an association between the mt4917A>G polymorphism belonging to the T haplogroup and AMD occurrence [86]. This polymorphism is located in the genes encoding NADH dehydrogenase, so the authors concluded that the association they observed was characterized by increased ROS production. A similar conclusion was drawn by SanGiovanni et al., who observed an association between AMD occurrence and the mtA11812A>G polymorphism in the NADH ubiquinone oxidoreductase gene [87]. The protective effect of the H haplogroup against AMD and a higher risk associated with the J haplogroup were confirmed by Kenney et al. [88]. To investigate the effects of different haplogroups, these authors created ARPE-19-based cybrids containing identical nuclei but different mtDNA variants. They observed a difference in the energetic profile between H and J haplogroups and concluded that this might underline the mtDNA-nDNA interaction, resulting in a change in the expression of seven mitochondrial and eight nuclear genes. The latter included genes encoding proteins of alternative complement, inflammation, and apoptotic pathways, with the potential to play an important role in AMD pathogenesis. Transmitochondrial cell hybrids were also used by these authors to demonstrate that cybrids having AMD mitochondria displayed reduced viability, decreased number of mtDNA copies, downregulated genes involved in metabolism of mtDNA and antioxidant defense. They are implicated in apoptosis, autophagy, and ER stress as well as doing more damage to mtDNA [89]. The general conclusion on the protective effect of H and increasing risk of J was confirmed in another study with 200 wet AMD Austrian patients [90]. However, such association was not confirmed in a large French cohort (1224 wet AMD patients and 559 individuals with normal fundus) [91].

Ballinger et al. observed damage to mtDNA in human transformed RPE cells exposed to hydrogen peroxide [92]. That damage was not completely repaired after a 3 h repair incubation. Moreover, these authors did not observe any DNA damage in three nuclear loci they investigated, and based on that observation, they concluded that there was preferential damage to mtDNA under hydrogen peroxide treatment. Furthermore, these authors concluded that there was a decreased redox function in RPE cells on hydrogen peroxide treatment on the basis of MTT reduction. However, these conclusions should not be generalized, as other works have shown that RPE cells, including ARPE-19 cell line, are prone to H_2_O_2_-iduced damage to their nuclear DNA [93].

Karunadharma et al. compared mtDNA damage typical for aging with that for AMD in human donor eyes obtained from an eye bank [94]. They noted that normal aging was associated with an increase in the common deletion region in mtDNA, but AMD was linked with elevated levels of mtDNA damage as compared with age-matched subjects without AMD. Based on the analysis of two nuclear genes, the authors noticed that mtDNA accumulated about eight times more damage than its nuclear counterpart. These authors concluded that damage in mtDNA may be an important element of AMD pathogenesis, as they may underline RPE dysfunction, crucial for AMD.

Godley et al. showed that mitochondria isolated from primary human RPE cells exposed to blue light produced hydroxyl radical, superoxide and singlet oxygen [95]. As a result, these authors observed an increased mtDNA damage in RPE cells exposed to blue light as compared with cells in the dark. Study with antioxidant suggested that superoxide anion might be primarily responsible for the observed mtDNA damage. That exposure was also associated with a small loss of mitochondrial activity. Therefore, blue light, a risk factor for AMD, may contribute to its pathogenesis through the induction of mtROS and mtDNA damage. Similar to the study of Ballinger et al., these authors did not observe damage to nuclear DNA, which was assessed by qPCR. The use of a small set of low molecular weight antioxidants to draw a definite conclusion on the involvement of a particular ROS in observed effects is somewhat uncertain due to the relatively low specificity of these compounds. Moreover, these authors did not present unequivocal evidence of a causative relationship between antioxidant-induced ROS scavenging and an observed decrease in mtDNA damage. The lack of nDNA damage cannot be generalized due to the experimental technique employed and data obtained in other laboratories.

These and other studies showing a greater extent of specific damage to mtDNA in the retinal cells than to nDNA usually do not contain a comparative analysis of other tissues. It is known that the rate of mutations in mtDNA is about an order higher than in nDNA. These mutations are preceded by DNA damages, which, in general, are less efficiently repaired in mitochondria than in the nucleus. Therefore, it is not surprising that the extent of DNA lesion in the retina observed in mtDNA is greater than in its nuclear counterpart. However, this may be different in other tissues and organs, and it is important to assess it in the retina in relation to other sites. This problem was addressed by Kenney et al., who showed a higher number of DNA rearrangements and deletions in mtDNA in retinas of AMD patients than non-AMD subjects [96]. Moreover, these authors observed a higher number of mtDNA changes in neural retina than in blood for both AMD patients and controls. They reported previously that the D-loop in AMD retinas had more genetic variations than in normal subjects [85]. The D-loop contains all three promotors for transcription of mitochondrial genes and the origin of replication of the heavy strand. Therefore, these results may suggest that disturbance in the replication and transcription of mtDNA caused by its damage/variability may contribute to AMD pathogenesis. However, these studies included a relatively small number of retina samples (13), which were compared with a higher number of blood samples (133 and 138). Moreover, retina samples taken postmortem were compared with blood samples obtained from live subjects.

There is a great difference between the structure and size of mtDNA and nDNA, resulting in a lack of a reliable technique to measure DNA damage in both organelles. To evaluate damage to mtDNA, a quantitative real-time polymerase chain reaction (qRT-PCR) is performed of two mtDNA fragments, one “long” (about 1000 bp) and the other “short” (up to 100 bp), and the ratio of DNA damage in these fragments is calculated [97]. Several fragments can be evaluated in this way, covering a significant portion of the entire mtDNA. This is practically impossible for nDNA, where qRT-PCR can be applied to measure DNA damage in specific regions of genes rather than to provide information on gross genomic DNA damage. Only damage, which stops DNA polymerase, can be detected in this way and, somewhat paradoxically, 8-oxo-G and some other small oxidative modifications to DNA bases do not stop DNA synthesis. It is out of the scope of this review to discuss all problems associated with the measurement of DNA damage in the nucleus and mitochondria, but all analyses comparing the extent of DNA damage in these two organelles should be considered skeptically.

Lin et al. showed that the extent of mtDNA damage in RPE cells was positively correlated with age of eye donors [98]. However, the repair capacity was inversely correlated with donors’ age. Furthermore, damages to mtDNA were preferentially located in the macular region rather than the periphery. A positive correlation was observed between the extent of mtDNA damage and AMD grading, which in turn was negatively correlated with DNA repair capacity.

As stated above, several attempts have been made to address the question as to why the central retina is particularly susceptible to initial factors of AMD pathogenesis. Terluk et al. investigated distribution of mtDNA damage in various regions of RPE and neural retina taken from AMD subjects [99]. They observed that mtDNA damage was limited to RPE, where its distribution did not differ between the macula and peripheral regions. These results are not in line with those obtained by Lin et al. [98]. Moreover, Terluk et al. concluded that damages to mtDNA were localized in the mitochondrial genome regions that may affect mitochondrial functions. In fact, due to almost complete packaging of mtDNA with coding sequences, it is hard to find its regions without potential consequences for mitochondrial function.

Wang et al. observed an increased concentration of 8-oxoG in aged rodent RPE and choroid and higher levels of DNA damage in mitochondria than the nucleus [100]. These authors found a decreased mRNA expression of some DNA repair enzymes in aged RPE and choroid, but this relationship was confirmed on protein levels for 8-oxoguanine-DNA glycosylase 1 (OGG1) and mutY homolog (MYH), which are primarily involved in removing oxidative DNA damage in the nucleus. In their subsequent work, these authors confirmed preferential damage to mtDNA and a decrease in the expression of some DNA repair enzymes also in the neural retina—photoreceptors and retinal ganglion cells [101].

Almost all mitochondrial proteins, including all DNA repair proteins, are encoded by nuclear genes. Therefore, it is clear that changes in nDNA may be reflected in mtDDR. The reverse relationship could also be considered a mutation in mtDNA, leading to ROS overproduction and damage to nDNA. Is this of any relevance to AMD? The problem was addressed by Miceli and Jazwinski, who showed that ARPE-19 cells deprived of their mitochondria changed the pattern of expression of some nuclear genes in a way similar to that observed in AMD [102]. That interesting study has several drawbacks, however—first, there was no control group, which is important because ethidium bromide, which has been used to eliminate mtDNA, has a strong affinity with double-strand DNA and could intercalate nDNA and affect expression of nuclear genes. Secondly, cells without mitochondria are deprived of a main source of energy, which is not always the case in mitochondrial dysfunctions, which in turn is apparently reflected in expression of nuclear genes. Importantly, both DNA intercalation and energy deficit should rather not result in any preference in affecting expression of particular genes.

## 7. Conclusions and Perspectives

In this review we showed that mitochondria might be a serious, if not the most serious, source of ROS playing a major role in AMD pathogenesis. We also discussed the main role of damage to mtDNA in the production of ROS.

At least two hypotheses supporting the central role of mtROS and damage to mtDNA in AMD pathogenesis can be considered (Figure 3). One is based on the assumption that initial factor is a ROS-inducing agent, the other that it damages mitochondria in a ROS-independent mode. Both scenarios lead to mitochondrial dysfunction that results in ROS overproduction and vicious cycle, leading to energy deficit and disturbances in cell functions, including such basic processes as replication and transcription. If cellular defense systems, which are also affected by increased ROS concentration, are not able to counterbalance detrimental changes resulting from mitochondrial dysfunction, the cell may accumulate pathological changes and ultimately die. However, the initial portion of ROS may also induce an adaptive response to oxidative stress, and a cell may further function in a pathological state. Such a cell is likely senescent, as senescence, and not cell death, could be primarily associated with RPE degeneration observed in AMD [34]. Therefore, the role of mitochondrial dysfunction in stress-induced senescence needs further studies. PGC-1α is potentially an important element at the crossroad of all these pathways—oxidative stress, mitochondrial dysfunction, mitochondrial defense, and senescence [103]. Satish et al. showed that activation of PGC-1α in ARPE-19 cells resulted in upregulation of mitochondrial genes and enhanced mitochondrial function in RPE cells by increasing basal and maximal respiration rates [104]. Another possibility is that some cells are intrinsically resistant to this stress. Moreover, mtDNA can also be considered a central player in AMD pathogenesis, as it can be a primary target for an initial factor, independently of whether it directly generates ROS or not. Damaged mtDNA likely results in disturbances in mitochondrial gene expression, as mtDNA damage may occur in either a coding sequence or a regulatory sequence.

As we stated above, no adult stem cells have been identified in the human retina. However, Salero et al. identified a subpopulation of human RPE cells that, after exposure to growth factors, displayed properties of adult stem cells [105]. Therefore, an alternative mechanism of regeneration of damaged RPE, and thus an alternative mechanism of AMD pathogenesis, can be considered. Consequently, mitochondrial metabolism in the RPE cells subpopulation indicated by Salero et al. should be investigated.

Study of the molecular aspects of AMD pathogenesis suffers from at least two disadvantages. The first is the lack of possibility of conducting research on target tissue in live subjects. The other is the lack of a reliable animal model of AMD, as this disease has significantly different phenotype in rodents than in humans. Several cellular models have been proposed, including cells taken postmortem from retinas of AMD patients. Recently, we have proposed an AMD model with cells with double knockout in the *NRF2* (nuclear factor-erythroid 2-related factor-2) and *PGC-1a* genes [106]. Earlier, Zhang et al. showed that mice with repressed PGC-1α fed with a high-fat diet provide a promising model to study AMD pathogenesis [107]. The RPE of these mice displays several abnormalities, including decreased mitochondrial activity and increased levels of ROS [107]. This model can be enriched by RPE cells obtained from human induced pluripotent stem cells taken from AMD patients with a genetic susceptibility to this disease, as shown by Golestaneh et al., who also noted an important role of PGC-1α in AMD pathogenesis [108]. However, Saint-Geniez et al. showed that PGC-1α regulates VEGF in the retina and is required for normal and pathological neovascularization [109]. This important work confirms a significant role of PGC-1α in AMD pathogenesis and shows that a tight regulation of this gene is crucial for retinal health and function.

When antioxidants are used, it should not to be assumed that their distribution inside the cell is homogenous, as mitochondria may have their own specificity to accommodate antioxidants [110]. This could explain discrepancies obtained in some clinical trials showing the influence of antioxidant supplementation on AMD occurrence and progression [111]. There is a great diversity among results on mtDNA oxidation reported in various laboratories—the results from different groups can differ by more than 60,000-fold [55].

Many conclusions and hypotheses on the central role of mitochondrial ROS and damage to mtDNA in AMD pathogenesis are based on the mitochondrial vicious cycle mechanism. This mechanism was also implicated in the free radical theory of aging [112]. As aging is the most serious risk factor of AMD, this implication may be of great significance for AMD pathogenesis. Contemporary considerations on the nature of aging assume rather that the main sources of mtDNA mutations are errors in mtDNA replication that escaped repair mechanisms due to their inefficiency (reviewed in reference [46]). Accumulation of such mutations in aging organisms occurs by clonal expansion and not the vicious cycle, and ROS play mainly a signaling role mediating response to accumulated damages to biomolecules associated with aging. Therefore, it may be important to specify the role of “physiological” and accelerated aging in AMD pathogenesis. One way or another, mitochondria and mtDNA are central elements of contemporary theories of aging, so they are central elements of AMD pathogenesis.

## Figures and Tables

**Figure 1 ijms-20-02374-f001:**
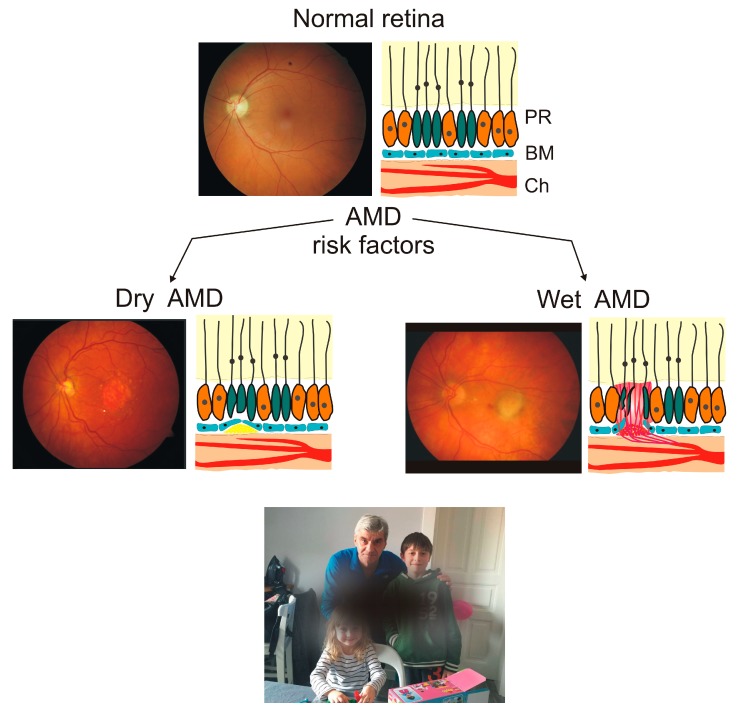
Age-related macular degeneration (AMD) is an eye disease affecting the macula, a central region in the retina. Presented are color pictures of fundus for normal retina and retina with changes typical for dry and wet AMD, two of its basic, clinically distinguished categories. Dry AMD is typified by the presence of drusen, yellowish objects between choroid (Ch) and Bruch’s membrane (BM), and photoreceptor (PR) loss. Wet AMD is associated with abnormal angiogenesis (choroidal neovascularization), leading to bleeding resulting in lifting up the macula from its normal position. Individuals affected by AMD in its advanced stage may experience problems with central vision.

**Figure 2 ijms-20-02374-f002:**
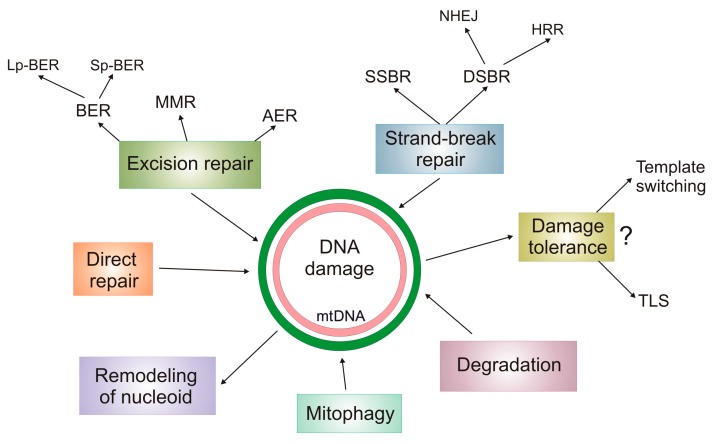
DNA damage response in mitochondria (mtDDR). In general, DNA damage in mitochondrial DNA (mtDNA) can be repaired or tolerated. Highly damaged mtDNA can be degraded, and mitophagy can contribute to this process although its exact nature is unknown. BER—base excision repair; Lp and Sp—long and short patch, respectively; MMR—mismatch repair; AER—alternative excision repair; SSBR and DSBR—single- and double-strand break repair, respectively; NHEJ—non-homologous end joining; HRR—homologous recombination repair; TLS—translesion synthesis. Remodeling of mitochondrial nucleoid has not been shown as a mtDDR pathway, but it can be assumed that it occurs if needed. The mechanism of DNA damage tolerance is hardly known in mtDNA, symbolized by a question mark.

**Figure 3 ijms-20-02374-f003:**
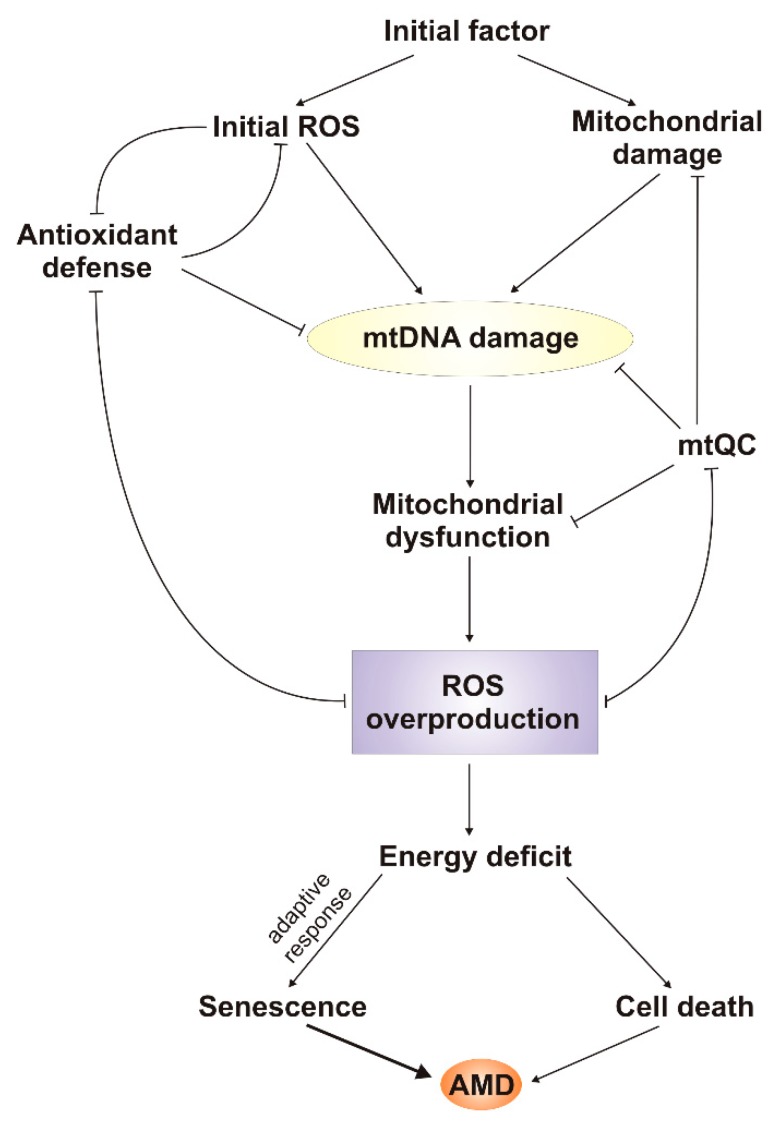
Possible involvement of mitochondrial reactive oxygen species (mtROS) and damage to mtDNA in the pathogenesis of AMD. Initial factor, which can belong to known AMD risk factors, may induce production of reactive oxygen species (ROS) or damage mitochondria. Both possibilities may result in damage to mtDNA, which in turn may initiate the mitochondrial vicious cycle, leading to energy deficit and ultimately to cell death. Initially and finally produced ROS can be neutralized by the cellular antioxidant system, and mitochondrial dysfunction in general and mtDNA damage in particular can be ameliorated by mitochondrial quality control (mtQC), but both are repressed by ROS production. Some cells may adapt to stress conditions and survive it, but they may display stress-induced premature senescence, leading to an inability to replace degenerated cells.

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
