# Peer review of "Role of Mitochondrial DNA Damage in ROS-Mediated Pathogenesis of Age-Related Macular Degeneration (AMD)"

_ijms, 2019, doi:10.3390/ijms20102374_

Round 1

Reviewer 1 Report

The review by Kaarniranta et al., describes the role of mitochondrial DNA damage in age-related macular degeneration.

The review has gathered recent publications on mitochondrial function, ROS generation, DNA damage response in mtDNA, mtDNA damage and repair in AMD and the role that mitochondrial dysfunction could play in the pathology of AMD.

The review is well written and is explicit. However, several recent references are left out in this manuscript.

1)    Page4, the authors mention that retinal pigment epithelial cells do not proliferate and adult stem cells have not been identified in the RPE. The authors have left out the paper published by Salero E, PMID: 22226358 in 2012. The authors need to at least cite this reference and discuss their findings.

2)    Page5, The authors write a paragraph about the paper published by Ferrington et al., in 2017 regarding the AMD RPE that reported the RPE of AMD donors were resistant to oxidative stress after 24h incubation with oxidative stress. However, they left out a paper that was published prior to this manuscript: PMID: 28055007, which showed that the susceptibility of AMD RPE to oxidative stress did not differ from normal RPE after 24h of incubation with hydrogen peroxide but after 48h of incubation the AMD RPE were more susceptible to oxidative stress-induced cells death. This paper also showed that mitochondria of RPE AMD were dysfunctional producing lower levels of ATP, whereas the ATP produced by glycolysis was higher in AMD RPE, indicating that ATP was essentially produced by glycolysis rather than mitochondrial activity in AMD RE.    

3)    Page 12, “ In this review we have shown ,” should be “In this review we describe that ….

4)    Page13, Here again the authors have not cited important papers that were published prior to their own publication: PMID: 26962700; PMID: 30524663; PMID: 23141926; PMID: 27998274, PMID:29925537

Author Response

Comment: 1)    Page4, the authors mention that retinal pigment epithelial cells do not proliferate and adult stem cells have not been identified in the RPE. The authors have left out the paper published by Salero E, PMID: 22226358 in 2012. The authors need to at least cite this reference and discuss their findings.

Answer: We have added the following fragment to the “Conclusions and perspectives” section:

As we stated above, no adult stem cells have been identified in the human retina. However, Salero et al. identified a subpopulation of human RPE cells that after exposure to growth factors displayed properties of adult stem cells [105]. Therefore, an alternative mechanism of regeneration of damaged RPE and thus an alternative mechanism of AMD pathogenesis can be considered. Consequently, mitochondrial metabolism in the RPE cells subpopulation indicated by Salero should be investigated.”

105. Salero, E.; Blenkinsop, T. A.; Corneo, B.; Harris, A.; Rabin, D.; Stern, J. H.; Temple, S., Adult human RPE can be activated into a multipotent stem cell that produces mesenchymal derivatives. Cell Stem Cell 2012, 10, (1), 88-95.

Comment: 2)    Page5, The authors write a paragraph about the paper published by Ferrington et al., in 2017 regarding the AMD RPE that reported the RPE of AMD donors were resistant to oxidative stress after 24h incubation with oxidative stress. However, they left out a paper that was published prior to this manuscript: PMID: 28055007, which showed that the susceptibility of AMD RPE to oxidative stress did not differ from normal RPE after 24h of incubation with hydrogen peroxide but after 48h of incubation the AMD RPE were more susceptible to oxidative stress-induced cells death. This paper also showed that mitochondria of RPE AMD were dysfunctional producing lower levels of ATP, whereas the ATP produced by glycolysis was higher in AMD RPE, indicating that ATP was essentially produced by glycolysis rather than mitochondrial activity in AMD RE.    

Answer: We have added the following fragment to the very end of the 3rd section:

“Prior to Ferrington’s paper, Golestaneh et al. showed that the susceptibility of cultured human RPE cells obtained from AMD donors did not differ from cells obtained from donors without AMD after a 24 h incubation with hydrogen peroxide, but after 48 h incubation, RPE cells from AMD donors were more susceptible to oxidative stress-induced cell death [42]. These authors also showed that mitochondria of RPE cells from AMD donors were dysfunctional, producing lower levels of ATP, whereas the ATP produced by glycolysis was higher, suggesting that ATP was essentially produced by glycolysis rather than mitochondrial activity, further supporting the hypothesis of the significance of energy crisis allocated to mitochondria in AMD pathogenesis.

42. Golestaneh, N.; Chu, Y.; Xiao, Y. Y.; Stoleru, G. L.; Theos, A. C., Dysfunctional autophagy in RPE, a contributing factor in age-related macular degeneration. Cell Death Dis. 2017, 8, (1), e2537.

Comment: 3)    Page 12, “ In this review we have shown ,” should be “In this review we describe that ….

Answer: We have changed that accordingly.

Comment: 4)    Page13, Here again the authors have not cited important papers that were published prior to their own publication: PMID: 26962700; PMID: 30524663; PMID: 23141926; PMID: 27998274, PMID:29925537

Answer: PMID: 26962700 (Iacoveli et al. 2016) was cited in our original submission (ref. no 41) and is in the revised manuscript.

PMID: 30524663 (Satish et al. 2018) – we have added the following sentence in the “Conclusion and perspectives section”:

“Satish et al. showed that activation of PGC-1a in ARPE-19 cells resulted in upregulation of mitochondrial genes and enhanced mitochondrial function in RPE cells by increasing basal and maximal respiration rates [104].”

104. Satish, S.; Philipose, H.; Rosales, M. A. B.; Saint-Geniez, M., Pharmaceutical Induction of PGC-1alpha Promotes Retinal Pigment Epithelial Cell Metabolism and Protects against Oxidative Damage. Oxid. Med. Cell. Longev. 2018, 2018, 9248640.

PMID: 23141926 (Saint-Geniez et al. 2013) – we have added the following sentence in the “Conclusion and perspectives section”:

“However, Saint-Geniez et al. showed that PGC-1a regulates VEGFA in the retina and is required for normal and pathological neovascularization [109]. This important work confirms a significant role of PGC-1a in AMD pathogenesis and shows that its stimulation may lead to pathological neovascularization.”

109. Saint-Geniez, M.; Jiang, A.; Abend, S.; Liu, L.; Sweigard, H.; Connor, K. M.; Arany, Z., PGC-1alpha regulates normal and pathological angiogenesis in the retina. Am. J. Pathol. 2013, 182, (1), 255-65.

PMID: 27998274 (Golestaneh et al. 2016) – we have added the following fragment:

“as it was shown by Golestaneh et al. who also noted an important role of PGC-1a in AMD pathogenesis [110]. “

as a continuation of our sentence “This model can be enriched by RPE cells obtained from human induced pluripotent stem cells taken from AMD patients with a genetic susceptibility to this disease.”

110. Golestaneh, N.; Chu, Y.; Cheng, S. K.; Cao, H.; Poliakov, E.; Berinstein, D. M., Repressed SIRT1/PGC-1alpha pathway and mitochondrial disintegration in iPSC-derived RPE disease model of age-related macular degeneration. J. Transl. Med. 2016, 14, (1), 344.

PMID:29925537 (Zhang 2918) – w have added the following fragment to the “Conclusion and perspectives section”:

“This was recently confirmed by Zhang et al. who has shown that mice with repressed PGC-1a fed with high-fat diet provide a promising model to study AMD pathogenesis [108]. RPE of these mice display several abnormalities, including decreased mitochondrial activity and increased levels of ROS.

Reviewer 2 Report

The manuscript „Role of mitochondrial DNA damage in ROS-mediated pathogenesis of age-related macular degeneration“ by Kaarniranta et al. aims to review recent findings on the relevance of reactive oxygen species and mitochondrial DNA damage in AMD pathogenesis.

While the authors address a point which is certainly of great interest and importance in the field of AMD research, unfortunately the manuscript fails to meet standards required for reviewing this topic. Essentially, the manuscript needs major revision in English grammar and vocabulary and would clearly benefit from a revision by a native speaker. Apart from numerous minor spelling and grammar mistakes there are also several passages in the text which reveal major deficits in English language to a point where the author’s statements cannot be appropriately followed due of greatly flawed language.

A second major flaw of the manuscript resides in textual inadequacies. Often, the contents presented are hard to follow, as there is no clear story line or logical order of thoughts in which data are presented.

Further, the manuscript describes a number of general information, e.g. on mitochondrial DNA, which is clearly covered in standard text books and does not add any relevant information with regard to AMD pathology.

Taken together, due to its poor quality this manuscript greatly fails to provide a valuable source of condensed information on mitochondrial DNA damage in the pathogenesis of AMD etiology.

Author Response

The revised version of the manuscript has been finally edited by the MDPI language service. We have removed Figure 2 (human mitochondrial DNA) and done our best to improve logic and consistency of that manuscript.

Reviewer 3 Report

In this review by Kaarniranta et al., the authors focus on AMD in terms of mitochondrial DNA damage and ROS related pathogenesis of this eye diseases. The review is generally well written and easy to follow. I have marked some minor points that need to be taken into account.

1)      What has happen with the “old” Fig.2 (line 219-220)? Is it still in the review, because the Figure legend is striked out within the text. However, I would recommend to take it out, because it does not fit into the context.

2)      Line 245: what kind of human diseases? Name a few.

3)      Line 424: use “studies” instead of “papers”

4)      Check the review carefully. There are some typos in the text.

Author Response

In this review by Kaarniranta et al., the authors focus on AMD in terms of mitochondrial DNA damage and ROS related pathogenesis of this eye diseases. The review is generally well written and easy to follow. I have marked some minor points that need to be taken into account.

Comment 1)      What has happen with the “old” Fig.2 (line 219-220)? Is it still in the review, because the Figure legend is striked out within the text. However, I would recommend to take it out, because it does not fit into the context.

Answer “Old” Figure 2 along with its legend have been removed from the manuscript according to suggestions of Referee #2.

Comment 2)      Line 245: what kind of human diseases? Name a few.

Answer We have changed the sentence

“There are many deletions and point mutations in mtDNA and some of them are associated with serious human diseases [52,53].”

into

“There are many deletions and point mutations in mtDNA and some of them are associated with serious human disorders, such as opthalmoplegia, migraine, dysphagia, sensorineural hearing loss, cognitive decline and others [52,53].”

Comment 3)      Line 424: use “studies” instead of “papers”

Answer We have changed accordingly.

Comment 4)      Check the review carefully. There are some typos in the text.

Answer We have double-checked the manuscript to correct typos.

Round 2

Reviewer 1 Report

The authors have addresses most of the reviewer’s comments, however, some statements are still misleading and need to be rectified.

1)    Page 14: Lane 535: “Several cellular models have been proposed including that of Ferrington…. “the paper PMID 28055007 has also proposed the AMD RPE cells from cadaver eyes and was published prior to Ferrington paper and should be cited in this sentence.

2)    Page 14: Lane 537: “Recently, we have proposed an AMD model with cells with double knockout in the NRF2 (nuclear factor-erythroid 2-related factor-2) and PGC-1a genes [1047]. This was recently confirmed by Zhang et al.,”

Again, this sentence is misleading! The paper by Zhang et al was published in 2018 and cannot be stated as “this was confirmed by Zhang et al.” Zhang et al should be stated before the NRF2 reference that was published later in 2019.

3)    Page 14: line 547: and shows that its stimulation may lead to pathological neovascularization….…” should be stated as “and shows that a tight regulation of this gene is crucial for retinal health and function”, (because reduced PGC-1a could also affect neovascularization in the retina.  

Author Response

The authors have addresses most of the reviewer’s comments, however, some statements are still misleading and need to be rectified.

Comment 1)    Page 14: Lane 535: “Several cellular models have been proposed including that of Ferrington…. “the paper PMID 28055007 has also proposed the AMD RPE cells from cadaver eyes and was published prior to Ferrington paper and should be cited in this sentence.

Answer The major problem in addressing this remarks is that we have received the communicate:

“The following term was not found in PubMed: PMID 26055007”

In fact, primary RPE cells obtained post-mortem, were used in several studies and we have mentioned Ferrington as she likely has done the largest work on mitochondria using such cells and we have cited her results in the present manuscript. We cannot see the need to cite all papers with RPE cells taken post-mortem from AMD subjects. Therefore we have changed the sentence

“Several cellular models have been proposed, including that of Ferrington and her coworkers, containing cells taken post mortem from retinas of AMD patients [40,106].”

into

“Several cellular models have been proposed, including cells taken post mortem from retinas of AMD patients.”

so we have removed any particular annotation and references, as no specific review has been published on that subject and too many original papers to be cited in this work.

Comment 2)    Page 14: Lane 537: “Recently, we have proposed an AMD model with cells with double knockout in the NRF2 (nuclear factor-erythroid 2-related factor-2) and PGC-1a genes [1047]. This was recently confirmed by Zhang et al.,”

Again, this sentence is misleading! The paper by Zhang et al was published in 2018 and cannot be stated as “this was confirmed by Zhang et al.” Zhang et al should be stated before the NRF2 reference that was published later in 2019.

Answer We have changed the sentence:

“This was recently confirmed by Zhang et al., who have shown that mice with repressed PGC-1a fed with high-fat diet provide a promising model to study AMD pathogenesis [108].”

Into

“Earlier Zhang et al. showed that mice with repressed PGC-1a fed with high-fat diet provide a promising model to study AMD pathogenesis [107].”

Comment 3)    Page 14: line 547: and shows that its stimulation may lead to pathological neovascularization….…” should be stated as “and shows that a tight regulation of this gene is crucial for retinal health and function”, (because reduced PGC-1a could also affect neovascularization in the retina. 

Answer We have changed the sentence

“This important work confirms a significant role of PGC-1a in AMD pathogenesis and shows that its stimulation may lead to pathological neovascularization.”

into

“This important work confirms a significant role of PGC-1a in AMD pathogenesis and shows that a tight regulation of this gene is crucial for retinal health and function.”